# A Review of Ampelometry: Morphometric Characterization of the Grape (*Vitis* spp.) Leaf

**DOI:** 10.3390/plants12030452

**Published:** 2023-01-18

**Authors:** Péter Bodor-Pesti, Dóra Taranyi, Tamás Deák, Diána Ágnes Nyitrainé Sárdy, Zsuzsanna Varga

**Affiliations:** 1Department of Viticulture, Institute for Viticulture and Oenology, Buda Campus, Hungarian University of Agriculture and Life Sciences, Villányi Str. 29-43., H-1118 Budapest, Hungary; 2Department of Oenology, Institute for Viticulture and Oenology, Buda Campus, Hungarian University of Agriculture and Life Sciences, Villányi Str. 29-43., H-1118 Budapest, Hungary

**Keywords:** grapevine, *Vitis*, leaf morphology, organometry

## Abstract

Grape (*Vitis* spp.) is one of the most important horticultural crops, cultivated worldwide on more than 7.3 million hectares for various purposes such as winemaking, fresh fruit consumption, rootstock, and ornamental plants. Based on the inter- and intraspecific morphological variability, several descriptor lists, manuals and ampelographic studies are available for identification. Among the organs, leaves have the most traits, while the young shoot, bunch and berry are also important in the characterization of the genotypes. *Vitis* species and cultivars are described by leaf morphological characterization developed in many ways for the identification of genotypes, to clarify synonymies and distinct clones or evaluate the diversity of wild *Vitis* taxa. Morphometric—also known as *ampelometric*—evaluation has an extensive background in the literature. However, for some reasons, only a part of the literature is cited, despite its significant scientific value. In this paper, we summarize the efforts of metric characterization of the grapevine leaf with the introduction of the scientific objectives and reviewing the studies showing the innovations in phenotyping during the past 120 years.

## 1. Introduction

Grapevine (*Vitis vinifera* L.) is one of the most important horticultural crops, the cultivation of which dates back to 5000–6000 BC in Transcaucasian regions [1]. Due to the large numbers of table grape and wine grape cultivars in production, in germplasm collections and in private gardens, detailed characterization has high importance for many reasons. First, there are variable cultivation demands of the genotypes produced around the world. Therefore, in quality-oriented production, it is essential to understand the phenological characteristics (time of the bud burst, véraison, ripening time), resistance/tolerance of the genotypes against pests or diseases, and the different vineyard practices (pruning and harvest criteria), which are detailed in many ampelographic albums [2,3]. Second, appellation requires that certain wine products (e.g., Csopak, Eger, Tokaj etc.) must be vinified from certain cultivars [4]. Third, the wine is usually varietally labeled, and this could have an impact on the pricing and consumer preference [5,6]; therefore, the use of the correct cultivar is of high importance.

According to the International Organization of Vine and Wine [7], to date, more than 150 descriptors are available for the characterization and identification of the genotypes. Among those, a range of morphological patterns, molecular genetic markers, and phenological perceptions can be found. Morphological observations are mainly based on the properties of the shoot, bud, leaf, bunch, berry, and seed, reflecting their qualitative (e.g., color, density, shape) and quantitative (size, number, weight) traits. Reproductive organs, i.e., flowers, bunches and berries, may also have valuable characteristics when used for identification, but their availability is limited to the vegetation period. Moreover, available traits used to compare the reproductive organs of the rootstock cultivars are also quite limited. For example, ‘Riparia portalis’, ‘Berlandieri×Riparia T5C’ and many other rootstocks have male flowers, and consequently, generative organs are of less importance from the aspect of characterization purposes. Wild *Vitis* taxa, such as the wild grape (*Vitis sylvestris* C.C. Gmel. Hegi), have dioecious flowers with functional male and functional female individuals; therefore, evaluation of the morphological diversity is most frequently based on the vegetative traits.

### Grapevine Leaf Morphological Traits

Grapevine leaves develop at the shoot tip, as it elongates. At each node, one leaf expands. The positioning of the leaves is distichous: leaves grow in two vertical rows along the shoots. The three distinct parts of each leaf consist of the petiole, bracts, and blade [8]. Lamina is formed by five principal veins (usually labelled as mid vein, N_1_; distal vein, N_2_; and proximal vein, N_3_) arising from the petiolar junction and ending at the tip of the lobes. Beside these veins, the venation network consists of secondary veins (for example petiolar vein, N_4_) and tertiary veins (for example, N_5_) [7]. From these veins, smaller veins are initiated and shape the blade. Depending on the cultivar, leaves could be entire or lobed (with 3, 5, 7 or 9 lobes), divided by the sinuses [1]. According to Mazade, the main features are the superior lateral lobe, superior lateral sinus, tooth, indentation, limb, inferior lateral sinus, inferior lateral lobe, inferior lateral lobe or terminal lobe, petiole or peduncle, petiolar sinus, secondary vein or sub-rib, mid-rib or principal vein, and margin (Figure 1) [9]. The size, shape, presence/absence, color, and further physical appearance of these features serve as the basis for leaf ampelographic characterization. Both young and adult leaves provide the possibility of phenotyping, but adult grapevine leaves carry the most descriptor traits compared to all other organs [7].

Leaf morphological traits can be described as qualitative and quantitative features. Qualitative characteristics are evaluated on an ordinal (e.g., density of the trichomes) or nominal scale (e.g., shape of leaf lamina, serrations, or sinuses). The description of these characteristics requires routine and standards for comparison. Quantitative traits can be measured on ordinal scales (e.g., numbers of lobes) or continuous scales (e.g., vein length). The evaluation of continuous angular and linear numerical traits is usually referred to as *ampelometry*. As far as we know, this expression was first mentioned in Louis Ravaz’s “*Les vignes americaines*” [10]. Later, it was explained in Seltensperger’s *Dictionnaire D’ Agriculture et de Viticulture* as “*Ampélométrie—Science qui a pour but la determination des varietés de vignes d’aprés la longueur relative des nervures principals de la feuille et les angles qu’elles formet entre ells*” (*Ampelometry—Science which aims to determine the grapevine varieties according to the relative length of the main veins of the leaf and the angles they form between them*) [11]. Kozma divided ampelometry into four sub-groups of study: *foliometry*, *florimetry*, *uvometry* and *carpometry* referring to the characterization of the leaf, flower, bunch and berry, and seed, respectively [12].

## 2. History and Development of Ampelometry

### 2.1. Prelude of Ampelometry

Although the expression *ampelometry* is only 120 years old, the bases of the study date back to the 16th century. Dodoens and Bauhin studied the leaf of the *Vitis vinifera* not only from viticultural aspects, but also from botanical aspects [13,14]. The expression “*ampelography,*” which refers to the detailed characterization of the *Vitis* genotypes, dates back to Phillip Jacob Sachs in 1661 [15]. Later, Rea and Worlidge studied grapevine cultivars according to the size and compactness of the bunch, lobature of the leaf and color of the berry [16,17]. Miller detailed 26 species according to their morphological traits, where properties of the leaves were also evaluated [18]. Duhamel studied eight species belonging to the *Vitis*, among which are *VITIS folii, laciniatis*. Cornu., *VITIS quinquefolia Canadensis Scandens*, Inst.*, VITIS Petroselini folio, Caroliniana*. These names refer to extreme leaf shapes and highlight the importance of this organ in classification [19]. Later, not only species, but also cultivars with interesting leaf shapes were included in botanical and viticultural books, such as the ‘Parsley grape’ which was mentioned as “*cioutat”* [20,21]. Clement classified cultivars based on the leaf lamina with the corresponding Latin expressions, such as very large (*folia maxima*), large (*maxima*), etc., and defined not only the size but also the lobature [22]. During the 19th century, many efforts were made to classify the large number of grapevine cultivars based on morphological traits (artificial classification). This was necessary due to the spread of synonym names (same genotype with different names) among cultivars. Among others, Vest [23], Acerbi [24], Gok [25], Babo and Metzger [26], Burger [27] and Dierbach [28] described and classified species and cultivars where the morphology of the leaf lobature and serrations were of high importance. In the middle of the 19th century, new pests and diseases appeared in Europe. Among those, phylloxera (*Daktulosphaira vitifoliae* Fitch.), downy mildew (*Plasmopara viticola* Berk and Curt.) and powdery mildew (*Erysiphe necator* Schwein.) were the most important. Against the aphid and the pathogens, new cultivars: rootstocks and direct producers, were introduced in the production, which needed ampelographic description. At this time, methods of the morphological characterization and classification showed rapid development, which are detailed in Molon’s outstanding work *Ampelografia: descrizione delle migliori varietà di viti per uve da vino, uve da tavola, porta-innesti e produttori diretti* [29].

### 2.2. Ampelometry by Ravaz

Traditional morphometry is based on linear and angular measurements of the organs, where data are evaluated with a multivariate statistical approach [30]. In ampelography, the bases of this study date back to the beginning of the 19th century, as earlier literature rarely quantified the morphological characters, but instead indicated leaves or bunches as “*small*” or “*large*”. Among others, Frege [31], Metzger [32] and Tersánczki [33] provided numerical data and a scale for the berry size and petiole length characterizations, and later, Goethe [34] suggested using the angle of the leaf petiole sinus in the ampelographic characterizations (for more details, see Branas [35]). Even though this method was novel in ampelography, it must be mentioned that the angles between the veins and numbers of lamina sections divided by the venation in many species, including grapevine, were in botanical studies investigated far earlier. “*So that where the fibres stand collateral with one in the middle, if you suppose them to be drawn out at opposite angles; or where the chief fibers part at the stalk, you only take in the stalk; you will thereby divide a circle into eight, twelve, or six equal parts; as in sirynga, the vine and others*” (Grew, [36]). In the beginning of the 20th century, Ravaz [10] improved the quantitative characterization describing the leaf venation and serration through metric observations, and, thus, the main components of his work were the evaluation of the angles between the veins (α, β, α + β) and ratios of vein lengths (Figure 2). Ampelometry traditionally provides linear and angular data; therefore, primary data is a set of continuous variables: the length of veins, the angles between the veins, the ratio of lengths, etc. Ravaz’s data evaluation was novel from both an ampelographic and statistical point of view as he provided ten categories for the sum of the angles between the veins (α + β), where the first category is ≤70°, the second is 71°–80° … and the tenth is 151°–160°. With this step, he facilitated the description and blurred intravarietal diversity. Ampelometry was later implemented by Seelinger [37], and an extended list of descriptors were applied to give a more detailed ampelometric description of rootstocks in Moog’s [38] work. According to Ravaz’s [10] study, French ampelographer Pierre Galet [39,40] made a comprehensive description of the grapevine cultivars based on linear and angular traits.

### 2.3. Galet’s Ampelometric Index

Another step in data evaluation was introduced by Galet [39,40], with the description of the ampelometric index. This evaluation does not compare the primary data resulting from the linear and angular measurements, but associates the ratios of certain traits followed by categorization of the data (Figure 3).

This index comprises the categorical evaluation of eight elements:L/W—length of the leaf/width of the leaf;N_2_/N_1_—length of the distal vein/length of the mid vein;N_3_/N_1_—length of the proximal vein/length of the mid vein;N_4_/N_1_—length of the petiolar vein/length of the main vein;Us/N_3_—depth of the upper sinus/length of the proximal vein;Ls/N_2_—depth of the lower sinus/length of the distal vein.

Additionally, the sum of the angles between the veins is measured as
α + β;α + β + γ;
where α is the angle between the mid vein (N_1_) and distal vein (N_2_), β is the angle between the distal vein (N_2_) and proximal vein (N_3_), and γ is the angle between the proximal vein (N_3_) and petiolar vein (N_4_). This method provides the possibility of evaluating the morphology and main shape of the leaves irrespective of their sizes, as ratios are not affected by increasing size. Galet [39] introduced several leaf forms and representative species, for example, cordiform (*Vitis cordifolia*, *Vitis cinerea*), cuneiform (*Vitis riparia*), truncate (*Vitis aestivalis*), orbicular (*Vitis vinifera* L.: ‘Chenin blanc’, ‘Carignane’, *Vitis Labrusca*) and reniform (*Vitis rupestris*), according to those typical ampelometric indexes with intermediate types, such as orbicular-reniform or cuneo-truncate (*Vitis monticola*, *Vitis candicans*). For more details, see Morton [41].

Continuous data of the ratios and sum of angles between the veins are turned into categorical variables which provide the ampelometric index (Figure 3). Galet [39,40,42,43,44], and later Németh [45,46,47] and Hajdu [48], provided the index for several wine grape, table grape and rootstock cultivars in their ampelometric albums, but for some reason, only a few research papers applied this method, while it has several advantages compared to the use of primary data. For example, Martí et al. [49] showed that most of the linear ampelometric features of ‘Parraleta’ and ‘Moristel’ varieties are significantly influenced by the year-to-year effect, while the angular traits and ratios of the linear features are less variable. Their results highlighted the differences among the cultivars as ‘Moristel’ was more influenced by the effect of the year as ‘Parraleta’ [49]. Recently, Chitwood reviewed Galet’s [39] ampelometry, and compared it with geometric morphometric methods. Based on the results, investigated samples were grouped into two discrete groups according to the sinus depth [50].

**Figure 3 plants-12-00452-f003:**
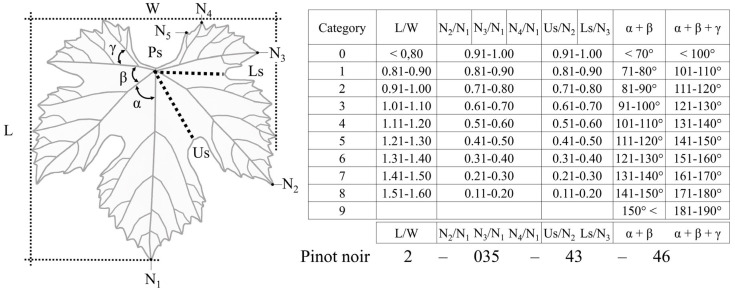
Ampelometric features and index according to Németh [51]. Nomenclature of the ampelometric traits is not uniform in the literature, for example, indication of the veins is different. In this study, we follow the OIV [7] descriptor list, where the veins are indicated with “N” which is identical to abbreviation “L” in Ravaz [10] and Galet [39], and “E” in Németh [51] numbered from 1 to 5.

### 2.4. OIV Descriptor List

The International Organization of Vine and Wine (Organization Internationale de la Vigne et du Vin—OIV) is one of the most important institutions in the viti-viniculture sector, providing statistical data about the World’s viticulture and oenology. They organize events and share standardized manuals for the description of grapevine genotypes. In 2009, OIV published the second Edition of the “OIV Descriptor List for Grape Varieties and Vitis Species” which contains more than 150 descriptor traits for the purposes of characterization and identification [7]. Among the morphological traits, quantitative (e.g., number of lobes, length of the bunch), qualitative (e.g., color, shape) and alternative (e.g., presence of teeth in the leaf sinus) characters are included, together with linear and angular ampelometric properties. Ravaz’s [10] measurements are the basis of the descriptors from OIV601 to OIV612, where length of the veins: N_1_ (OIV601), N_2_ (OIV602), N_3_ (OIV603), N_4_ (604); depth of the sinuses: length from the petiole sinus to the upper lateral leaf sinus (OIV605), length from petiole sinus to lower lateral leaf sinus (OIV606); angles between the veins: angle between N_1_ and N_2_ measured at the first ramification (OIV607); angle between N_2_ and N_3_ measured at the first ramification (OIV608); angle between N_3_ and N_4_ measured at the first ramification (OIV609). These traits are like those in Galet [39,40,42,43,44] and Németh [45,46,47], while further ampelometric features are also included in the descriptor list (Table 1).

In contrast to the ampelometric index where ten categories are provided, the OIV classified the values (vein lengths, sinus depths, etc.) into five categories. For example, the main vein (N_1_) is indicated as *very short* (up to about 75 mm), *short* (about 105 mm), *medium* (about 135 mm), *long* (about 165 mm) and *very long* (about 195 mm and more) based on 10 leaves (Table 1). From a statistical point of view, the 10 leaves could be evaluated individually, and the most frequent value (mode) would be typical to the genotype. The other possibility is to calculate the average value of the 10 leaves’ main vein lengths, and this value would be classified as *very short* or *short*, etc. As the data evaluation method is not defined by the OIV [7], it is helpful to include it in the interpretation of results of any study.

It must be highlighted that there is a similarly to several other species; grapevine (and in general the *Vitis* spp.) has more or less symmetric leaves with the same size of blades on each side of the mid vein. Consequently, ampelometric features (except for the length of the mid vein and the opening of the petiole sinus) are also symmetric, and one leaf sample provides two sets of data for each feature. In most studies, OIV categories are given as single values, even if the descriptor list suggests the observations on both halves of the leaves. There are only limited studies dealing with the symmetry of the lamina. Dorsey (1912) [52], for example, investigated the symmetry of the leaves concerning the angles between the veins (α and α + β). The reported data allow us to make further calculations. The angles of the two sides did not significantly differ from each other, while the linear Pearson correlation was low but significant (α: corr.: 031 *p* < 0.05; and α + β: corr.: 0.61 *p* < 0.01). This is in line with a later investigation of 22 bilateral traits of the ‘Pannónia kincse’ grapevine cultivar, where Bodor et al. [53] found a significant correlation between the two halves of the leaves (except for OIV608 where significance was not verified). Alba et al. [54] excluded those samples which were asymmetric in their study, as those were considered atypical. Abiri et al. [55] evaluated the two sides of the investigated samples separately. Moreover, a simple correlation among the traits allows the correlation between the morphological variables to be found.

**Table 1 plants-12-00452-t001:** Ampelometric characteristics in the OIV [7] descriptor list.

Code N°	Characteristic on the Mature Leaf	Notations				
		1	3	5	7	9
OIV 601	Length of vein N_1_	very short up to about 75 mm	short about 105 mm	medium about 135 mm	long about 165 mm	very long about 195 mm and more
OIV 602	Length of vein N_2_	very short up to about 65 mm	short about 85 mm	medium about 105 mm	long about 125 mm	very long about 145 mm and more
OIV 603	Length of vein N_3_	very short up to about 35 mm	short about 55 mm	medium about 75 mm	long about 95 mm	very long about 115 mm and more
OIV 604	Length of vein N_4_	very short up to about 15 mm	short about 25 mm	medium about 35 mm	long about 45 mm	very long about 55 mm and more
OIV 605	Length petiole sinus to upper lateral leaf sinus	very short up to about 30 mm	short about 50 mm	medium about 70 mm	long about 90 mm	very long about 110 mm and more
OIV 606	Length petiole sinus to lower lateral leaf sinus	very short up to about 30 mm	short about 45 mm	medium about 60 mm	long about 75 mm	very long about 90 mm and more
OIV 607	Angle between N_1_ and N_2_ measured at the first ramification	very small up to about 30°	small about 30°–45°	medium about 46°–55°	large about 56°–70°	very large about 70° and more
OIV 608	Angle between N_2_ and N_3_ measured at the first ramification	very small up to about 30°	small about 30°–45°	medium about 46°–55°	large about 56°–70°	very large about 70° and more
OIV 609	Angle between N_3_ and N_4_ measured at the first ramification	very small up to about 30°	small about 30°–45°	medium about 46°–55°	large about 56°–70°	very large about 70° and more
OIV 610	Angle between N_3_ and the tangent between petiole point and the tooth tip of N_5_	very small up to about 30°	small about 30°–45°	medium about 46°–55°	large about 56°–70°	very large about 70° and more
OIV 611	Length of vein N_5_	very short up to about 15 mm	short about 25 mm	medium about 35 mm	long about 45 mm	very long about 55 mm and more
OIV 612	Length of tooth of N_2_	very short up to about 6 mm	short about 10 mm	medium about 14 mm	long about 18 mm	very long about 22 mm and more
OIV 613	Width of tooth of N_2_	very short up to about 6 mm	short about 10 mm	medium about 14 mm	long about 18 mm	very long about 22 mm and more
OIV 614	Length of tooth of N_4_	very short up to about 6 mm	short about 10 mm	medium about 14 mm	long about 18 mm	very long about 22 mm and more
OIV 615	Width of tooth of N_4_	very short up to about 6 mm	short about 10 mm	medium about 14 mm	long about 18 mm	very long about 22 mm and more
OIV 616	Number of teeth between the tooth tip of N_2_ and the tooth tip of the first secondary vein of N_2_ including the limits	very small up to about 3	small about 4	medium about 5–6	large about 7–8	very large about 9 and more
OIV 617	Length between the tooth tip of N_2_ and the tooth tip of the first secondary vein of N_2_	very small up to about 30 mm	small about 30–45 mm	medium about 46–55 mm	large about 56–70 mm	very large about 70 mm and more
OIV 618	Opening/overlapping of the petiole sinus	wide open up to about −35 mm	open about −15 mm	closed about −5 mm	overlapping about 25 mm	very overlapping about 45 mm and more

### 2.5. Ampelometric Data as Continuos Variables

Instead of dividing the linear and angular continuous variables into categories, there is the possibility of evaluating the primary data of the ampelometric measurements. Mean and standard deviation supplemented with the coefficient of variation of the traits are valuable to show the typical leaf characteristics [56,57], while many authors compared primary data with ANOVA [58]. In the case of large numbers of variables, a possible method is principal component analysis (PCA) of the primary data, which was applied, for example, by Harib Ben Slimane et al. [59] and Labagnara et al. [60]. This method requires further analysis, as it was introduced by Alba et al. [54], who verified multivariate normality before running the PCA. Furthermore, as ampelometric traits are both linear and angular with different weight, they standardized the data to have a comparable scale. Discriminant analysis (DA) and stepwise discriminant analysis (STEPDISC) are also possible methods to find those characteristics that help discrimination and identification of the genotypes [61]. Clustering is a suitable method for the visual interpretation of the similarity/diversity between the samples according to the morphological traits [62]. Alba et al. applied chi-squared automatic interaction detection (CHAID) and classification trees to represent the relation between 100 table and wine grape genotypes. With this method, the authors were able to differentiate certain cultivars, for example ‘Italia’ and ‘Red Italia’, which was not possible on the basis of the six most frequently applied SSR loci [54].

### 2.6. Geometric Morphometry—Odontometry

In the late 1930s, Rodrigues introduced a novel method for leaf morphological characterizations, called *odontometry*. These investigations aimed to record biometric landmark positions along the outline of the leaf [63,64,65,66,67]. Analysis of the coordinates were carried out at three sections along the margin on the terminal lobe (from A to A_1_), lateral lobe (B’_1_ to b_1_) and superior lateral lobe (C’_1_ to c_1_) (Figure 4). Beside the average position and standard deviation of these landmarks, the numbers of teeth in each section were also evaluated. According to the investigations, he introduced the variability of the landmark point position of several rootstock cultivars and the difference among the leaves collected from different nodes of the shoot. This methodology did not spread widely among the ampelographers. Although some results were reported on the effect of the sampling years and growing locations on leaf morphometry, odontometric data were included in the characterization of grapevine genotypes [68,69,70].

### 2.7. Landmark-Based Geometric Morphometry

A possible method of morphological investigation is landmark-based geometric morphometry [71]. This study, similarly to Rodrigues’s [65] method, is based on the record of the location (x,y coordinates) of homologous biological landmarks [72]. In case of the grapevine leaf, there are numerous landmarks, and, hence, irrespective of the cultivar, leaves have a petiole junction point, five main veins that are divided into secondary veins. All main and secondary veins run into the serrations around the leaf lamina. Vein branches and tips are homologous points, perfect bases of the landmark-based geometric morphometric analysis. Besides the position of these points, the distance between them and the angle between the straight lines provide additional traditional ampelometric information. The definition of two landmarks provides one linear data set (length), while three landmarks result in three linear and three angular traits. If we increase the landmarks to 7, it will provide 21 linear and 105 angular traits, etc. (Figure 5), though an increased number of traits would result in a redundancy caused by the significant correlation between traits. Following the record of the landmarks for the GMM analysis, statistical evaluation is necessary. For this step, several tools are available, for example MorphoJ [73] and PAST [74], which proved to be powerful platforms for these evaluations. The most recent leaf morphological descriptions of the members of the *Vitis* taxa based on landmark-based geometric morphometry (GMM) were reported by Chitwood et al. [50,75,76,77], Bodor et al. [53,78] and Bryson et al. [79].

### 2.8. Elliptic Fourier Descriptors

While ampelometric traits are considered as traditional morphometric characters, other methods were also applied to characterize grapevine genotypes. Diaz et al. [80] and later Mancuso [81] published a new way of ampelometric investigations. In their studies, elliptic Fourier analysis (EFA) was carried out. This method is based on an outline analysis of closed contour objects. During this procedure, a chain code is obtained from the outline and normalized EFDs (elliptic Fourier descriptors) are calculated. With the help of the EFA, a representative shape of the grapevine leaf can be reconstructed, and variability is estimated according to PCA. Recently, Chitwood et al. [75] applied this method to investigate the morphological variability of grapevine cultivars and European wild *Vitis* taxa.

## 3. Tools for Ampelometric Evaluations

Dorsey [52] applied a transparent protractor to determine the angular dimensions of the leaf; however, from the previous time, we have little knowledge about the methodology of the measurements. Galet [40] (in Morton [41]) included a ruler and a protractor in his book, which was later also applied in other studies [82]. Boursiquot et al. [83] suggested that the large numbers of clones and cultivars with high variability needed computer-based data evaluation. As digital and computer-based image analysis developed, it was also implemented in ampelometry [84]. Martinez and Mantilla [85] for example, reported morphological investigations on the ‘Albariño’ grapevine cultivar, and carried out the leaf morphological analysis with a Digital Image Processing System MIP 1.4 (MICROM). Alessandri et al. [86] and Soldavini et al. [87] developed ampelometric computer software: AmpeloCADs and Superampelo. The latter was widely applied, for example, by Storchi et al. [88], Zdunić et al. [89] and Fatehi et al. [90]. In Martí et al. [49], the ampelometric evaluation was carried out with a Delta-T analyzer (Devices Ltd., Cambridge, UK) and DIAS software. Later, the GRA.LE.D. (GRApevine Leaf Digitalization) software was developed by Bodor et al. [91,92] to help the morphological investigation of the *Vitis* leaf according to the record of 30 landmarks and 2 semi-landmarks around the leaf boundary and branch-point of the venation, calculating more than 50 phyllometric traits. Chitwood et al. [76,77] used ImageJ [93] to place the investigated landmarks on the leaf samples.

Analysis of the leaf traits is not the single purpose of ampelometry; reconstruction of the leaf shape is also highly informative. Martinez and Grenan [94] gave one of the most spectacular graphic reconstructions of the leaf. After digitalization and digital measurements were carried out on each image, they graphically and manually reconstructed the representative leaf of the cultivars during a 12-step process. Later, Santiago et al. [95,96,97], Boso et al. [98], and Beleski and Nedelkovski [62] published comprehensive ampelometric descriptions based on this graphic reconstruction method. Outline analysis of the leaves, for example according to EDF, provided further possibilities to represent the average leaf form [81]. Evaluation of the geometric morphometric landmarks are also valuable as average coordinates of the landmarks give the possibility of graphic reconstruction (Figure 6).

## 4. Morphological Variability along the Shoot: Which Leaf to Compare?

As discussed above, certain morphological traits of the leaf are uniform among cultivars or even species, but surprisingly, there are notable differences even along the shoot [99]. The size and shape of the leaves differ along the shoot, e.g., basal, middle, and apical leaves are not completely similar. Those at the top of shoot are younger than those at the bottom; this is a reason for the shape and size difference, while the other reason is the order of each leaf along the shoot. This variability was already discovered in the 19th century. Demaria and Leardi [100] stated that the leaf would give the possibility for the organographic description, because it has some constant features. At the same time, they highlighted the importance of the sampling position. Leaves at the top and on the base of the shoot are not suitable for comparing the cultivars, in contrast to those that are in the middle of the shoot or close to the bunches. According to Ravaz [10], leaves from the 9th to the 12th nodes are the most similar, and thus, a comparison of these is recommended. This is in line with the recommendations of the OIV [7], suggesting the sampling from the middle third of several shoots. From an ampelometric point of view, recent investigations showed that the variability of morphometric traits and landmark positions around the lamina differ on each node. According to the record of 32 landmarks and 54 traits, the variability is lowest in the case of the leaves collected from the 11th node, while the largest is on the 7th node (Figure 7). Besides the traits, morphometric landmark positions are also different, as influenced by the node position (Figure 6) [78]. Cousins and Prins (2008) [101] emphasized that the transition of leaf forms along the shoot is often close to the distal inflorescence, which shows the link between this phenomenon and the GA gradient being responsible for the inflorescence development. Later studies explained the size and shape variability with allometry and heteroblasty, which are introduced in a more detailed fashion in Chitwood et al. [77,102] and Bryson et al. [79].

**Figure 6 plants-12-00452-f006:**
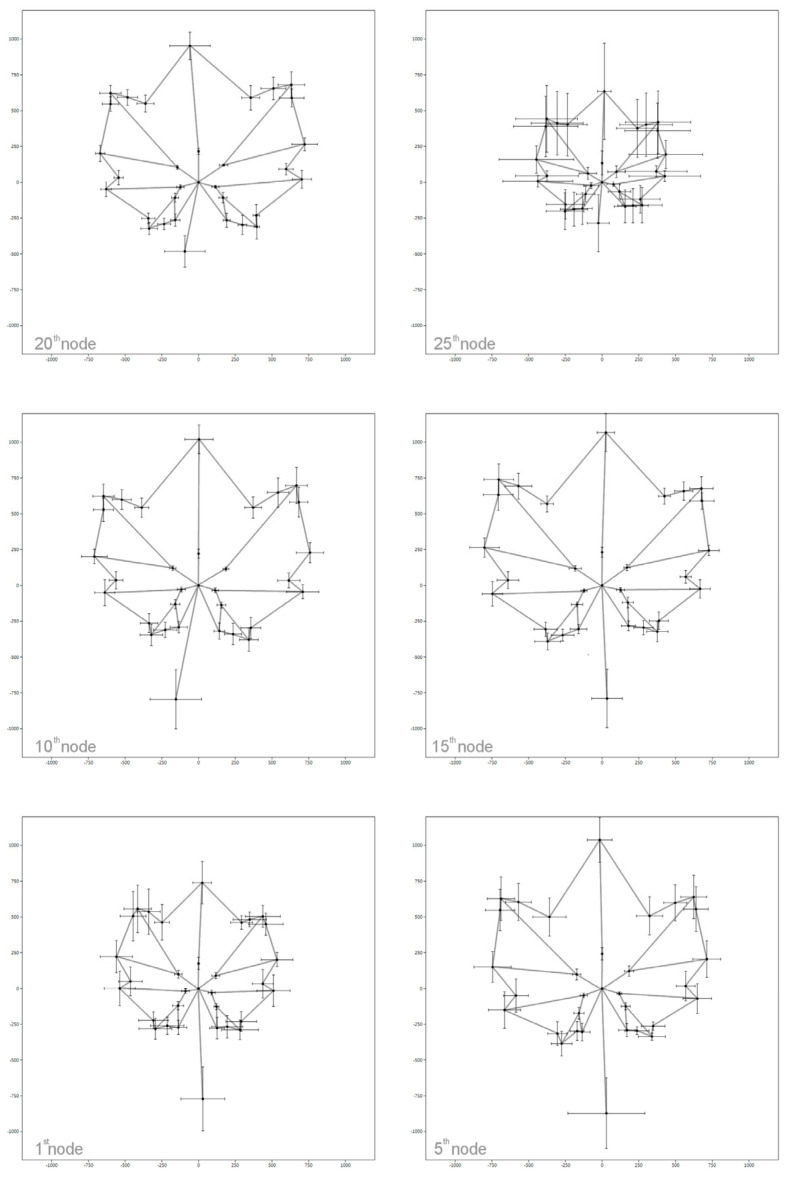
Mean and standard deviation of the x,y coordinates of 32 landmarks on the leaves along the shoot axis on the 1st, 5th, 10th, 15th, 20th, and 25th nodes (Bodor et. al, [78] with permission from the publisher).

## 5. Leaf Morphological Diversity among Species, Cultivars and Clones

The number of grape species, grapevine cultivars, rootstocks, clones and clone candidates range from 5000 to 22,000. This high number requires a detailed description from a botanical and viticultural point of view. Over the past 120 years, leaf morphometric characterizations were aimed at describing, for example, the diversity of the wild *Vitis* taxa, clearing synonymies of cultivar names, or distinguishing clones and clone candidates.

### 5.1. Ampelometric Evaluation of Vitis Species and Rootstock Cultivars

At the end of the 19th century, the spread of phylloxera forced European grapevine growers to use North American *Vitis* species (*Vitis riparia* Michx., *Vitis Berlandieri* Planch., etc.) and their cross-bred or selected cultivars (‘Berlandieri x Riparia T.K.5BB’, ‘Riparia portalis’, etc.) as rootstocks and direct producers (‘Noah’, ‘Elvira’, ‘Delaware’, etc.) as winegrapes. The high numbers of new cultivars required detailed characterization, and the inclusion of them in ampelographic albums and scientific studies. Bases of ampelometry date back to Ravaz [10] who introduced the phyllometric data of several species and also, for example, those cultivars: ‘Scuppernong’ (*Vitis rotundifolia*), ‘Early Victor’, ‘Martha’, ‘North America’, etc. (*Vitis labrusca*); ‘Riparia a Bourgeons Bronzés’, ‘R. Grand Glabre’, ‘R. Fabry’, ‘R Gaston Bazille’, etc. (*Vitis riparia*); ‘R. violet’, ‘R. Mission’, ‘R. de. Forthworth’, etc. (*Vitis rupestris*); ‘Æ sauvage’, ‘Æ de Spaunhorst’, etc. (*Vitis aestivalis*); and further important ones in the breeding history of the direct producers and rootstocks. Later, several authors reported the angular traits of the *Vitis* species and rootstock cultivars [38,39,42,52]. In the third volume of the *Ampelographic album*, Németh [47] introduced a detailed ampelography of direct producers and rootstocks widespread in Hungary. Among the traits, the author lists those leaf ampelometric characteristics which give the base of the ampelometric index: ratios of the vein lengths, and sinus depths, including the sum of angles. These albums did not investigate diversity or any external factors influencing the traits, but served as a reference for the identification of any unknown genotype. Among the first diversity investigations, Swanepoel and Villiers [82] aimed to compare the *Vitis* species, grapevine, and rootstock cultivars according to numerical taxonomy. Their results showed that cluster analysis performed on leaf ampelometric traits are able to distinguish American and European samples (with two exceptions being ‘Jacquez’ and ‘Muscat Alexandria’, since those were misclassified as European and American genotypes, respectively). Recent geometric morphometric investigations showed that composite leaf modelling is a more powerful method in the prediction of the species than the investigation of the individual leaves [79].

### 5.2. Morphometric Diversity of the Wild Grape (Vitis Sylvestris C.C. Gmel. Hegi)

Wild grape (*Vitis vinifera* subsp. *sylvestris* C.C. Gmel. Hegi) is an important resource both agrohistorically and genetically, and serves as a possible source of resistance. The description of the species dates back to Gmelin’s *Flora Badensis Alsatica* [103]. Since the phylloxera epidemic, intensive forestry and the appearance of the invasive North American *Vitis* species drifted wild grape to the edge of extinction in Europe; therefore, collection, description and preservation are important. According to Anzani et al. [104], wild grape has small leaves, a low length/width ratio and s wide petiolar sinus angle width. The lobature of the leaves varies from non-lobed to deep lobed and is different in Northern, Central and Southern Italy. Leaf morphology depends on the sex of the plant: male individuals have smaller leaves than females. Concerning the flower type and its influence on leaf morphology, Söylemezoglu et al. [105] found that wild grape samples had a broad variation in ampelometric traits even within the male and female individuals. Susaj et al. [106] also highlighted the variability of the traits within and between populations. In their study, samples were collected from three regions of Northern Albania, and leaves were investigated according to the OIV [7] descriptor list, and, for example, the length of the main vein showed no variability (OIV601—1) in one of the populations, while three classes of variability (OIV601—1, 3, 5) were observed in the other location. Martinez de Toda and Sancha [107] showed that, compared to the ‘Tempranillo’ and ‘Garnacha Tinta’ *Vitis vinifera* L. cultivars, wild grape individuals have smaller leaves, and a more open petiole sinus, while the lobature was found to be equally variable in both species. Cunha et al. [108] showed that morphological investigations are suitable to show the natural diversity of the species, and found that certain traits, such as the size of the leaf, the length of teeth compared with their width at the end of the base and the blistering of mature leaf, are discriminant characteristics of the populations. Slimane et al. [59] collected *Vitis sylvestris* ecotypes in the northwest region of Tunisia and compared the individuals according to 33 parameters. Different indices were calculated from their measurements, and then PCA was undertaken from the acquired data. Their study helped to identify the parameters that are most useful to distinguish the different ecotypes. In 2010, Ekhvaia and Akhalkatsi [109] conducted a detailed morphometric study of wild grapevine populations found in different parts of Georgia. They compared their leaf and flower morphology by methods of traditional and landmark-based geometric morphometrics. They found three phenetically distinct morphometric groups that were distinguished by the length of the main leaf veins (N_1_ and N_2_) and the length of the nectaries on male flowers. In the study of Barth et al. [110], 34 wild grapevine collections and a defined subpopulation from the Upper Rhine Valley were described and compared with six collections from the former Yugoslavia. The collections were described by ampelographic descriptors. Ampelographic traits split the collections into two major groups: one group mainly comprises samples from the Ketsch area and the Upper Rhine Valley. Bodor et al. [111] investigated the germplasm collection of the FEM-IASMA, and showed that samples with a different sex of the flower have different leaf morphology, and the year-to-year effect also modifies the ampelometric parameters.

### 5.3. Comparison of Grapevine Cultivars and Clones

The morphological identification of grapevine cultivars dates back to ancient times, and over the centuries, several hundreds of morphological traits were described to differentiate the genotypes. Several studies, for example Asensio et al. [112], Sabir et al. [113] and Popescu et al. [114], showed that some traits that are suitable for the comparisons.

Over the past 120 years, several ampelographic albums and research papers introduced discriminative ampelometric traits of cultivars, clones, and clone candidates. These morphological characteristics could help the evaluation of the morphological diversity, clear synonymies or serve genetic resource purposes. Ampelometric traits could be recorded by hand and digital image analysis. In the former case, the process is time consuming; therefore, the definition of those traits that are effective is of high importance. With the help of angular measurements, Averna-Saccá [115], cited in Allen [116], investigated the correlation between yield, sugar content, acidity and angle between the median vein and exterior lateral vein. Santiago et al. [95,96,97] showed that ampelometric investigations, combined with traditional ampelographic observations and molecular genetic analysis, are effective tools in the comparison of local cultivars, and they clear synonymies, or prove the existence of old cultivars and differentiate them from other genotypes. Statistical evaluation of 144 grape genotypes’ data included in Németh’s *Ampelographic albums* [45,46,47] showed that the traits have high discriminative power, as stepwise discriminant analysis correctly classified cultivars according to utilization (wine grape—97%, table grape—62%) and origin (convar. *pontica*—80%, convar. *orientalis*—66%, convar. *occidentalis*—36%) [117]. Preinier et al. [61] investigated 360 leaf samples from eleven cultivars and found seven parameters that provide a 100% correct classification of the samples. Abiri et al. [55] investigated 55 grapevine cultivars according to 33 morphological traits, including three linear and three angular ampelometric characters. Their results showed a low efficiency of the foliometric traits for the length of the main veins and main vein angle showed the lowest CVs (13.62% and 14.33%, respectively) among the investigated traits.

Clones are selected somatic mutants of grapevine cultivars; even the expression suggests genetic identity. There are minor genetic differences compared to the original cultivars mainly caused by bud mutations. These minor differences provide phenotypic variance, for example, in the vegetative performance, bunch architecture or must chemical compounds [118,119,120]. As these traits are linked to the coded region of the DNA, the most frequently applied molecular genetic markers, such as the SSR, are not always effective in clone identification and discrimination, as these markers amplify the non-coding region [121]. Therefore, morphological descriptions have a high importance [122]. Nieddu et al. [58] showed that many of the OIV descriptors were ineffective to differentiate several ‘Vermetino’ clones from each other, even if recorded on the mature leaf, for example, the shape of the blades, goffering of the blade, profile, shape of the teeth, degree of petiole sinus opening, shape of base of petiole sinus. In line with this, the main ampelometric traits (for example, OIV601, 602, 603, 605, 606, 608, 609, 612, etc.) also showed constant OIV categories. Differences among the clones were detected in the traits OIV 604, 607, 613, 079-1 and ratios of 605/066-4, 606/066-4, 066-5/066-4, 601/604, and those were significant. These results showed the effectiveness of both primary and calculated traits. Bodor et al. [53] also showed that ampelometric investigations according to the OIV system included Galet’s trait ratios and landmark-based geometric morphometry are powerful tools in the differentiation of clone candidates. While some of the OIV traits showed constant values (e.g., OIV601, 604), others were variable (OIV606, OIV603). In contrast with these findings, Galet’s calculated ratios and sum of angles between the veins were variable among the genotypes. Landmark-based geometric morphometry and discriminant analysis proved to be of low efficiency for correct classification.

Detailed morphological characterization has great importance in the identification and differentiation of not only natural, but also cross-bred cultivars, as those are a result of a long, expensive, time and labor consuming process. New cultivars are patented in many countries according to those origins (as well as the date and place of the cross): morphological traits, phenology, and resistance. Concerning the morphology, vines, canes, leaves, trunks, flowers, and the fruits are introduced according to qualitative and quantitative traits. Ampelometry plays an important role in the patents, for example of ‘Sweet scarlet’ and ‘Catena Malbec clone 13′ (United States Plant Patent: US PP15,891 P3 and US PP20,859 P3, respectively), as the length of the veins (L1, L2, L3) and the angle between the mid vein L1 and L3, and between L1 and the first vein of L3 are included.

## 6. Factors Influencing Ampelometric Traits

According to the different studies discussed above, there are noticeable differences among the *Vitis* species, grapevine cultivars, clones and even clone candidates, but there are other factors which have a significant effect on the leaf morphology, even within the same genotype. At the taxonomic level of the species, Anzani [104], and later some other studies [111], showed that ampelometric traits of the male and female individuals of the wild grape are different, which should be considered when estimating the natural diversity of the species. Further studies linked to the grapevine showed that morphometry is influenced by the vineyard maintenance. Silvestroni et al. [123] found that virus infection caused by grapevine leaf roll (GLR), grapevine fleck, grapevine stem pitting (LR), and grapevine fanleafvirus (GFV) would cause noticeable differences in the traits. Santos et al. [124], who performed multivariate discriminant analysis according to ampelometric traits of virus infected ‘Arinto’ samples, later verified this result. Classification of the GFKV (grapevine fleck virus), GFVL (grapevine fanleaf virus), GVLRV3 (grapevine leaf roll virus3) and non-infected samples showed a 74.2% correct classification, emphasizing that infections (except GFKV) cause typical leaf deformations.

Other authors [85,125] found that phyllometric traits are influenced by the origin of the plants, as ampelometric characters significantly differ between samples from micropropagated and vegetative propagated (woody shoot cutting) plants. Discriminant analysis showed a highly correct grouping, meaning that traits are typical within plants with the same propagation type [125]. Martinez et al. [126] investigated the ampelometric traits in a complex experiment, and found that years, soil types (poor sandy soil and richer silty soil), and origin (control clone and somaclone plants) would have a significant effect on both linear and angular traits, and ratios of the former characteristics. Among the traits, the least variability was observed in the case of angular traits. Carrión et al. [127] showed that leaves obtained from vines with different irrigation and conduction (vessel and trellis) systems differ regarding the leaf morphometric characters. Bodor et al. [128] investigated ‘Syrah’ and ‘Sauvignon blanc’ leaf samples collected from plots with different bud-loads. The results showed that the number of shoots and vegetative load have a significant effect, especially on the linear traits. Beside the vineyard operations, ecological factors and row orientation also have a significant effect on the leaves. Further investigations showed that ‘Furmint’ leaf morphometry is altered by the elevation of the plots and the orientation of the rows [92]. These results highlight the importance of a detailed interpretation of sampling.

## 7. Discussion

Chadha and Randhawa [99] and citations therein showed the importance of leaf morphological investigations and emphasized that grapevine leaf characteristics without the observation of other organs would be sufficient for the classification of grapevine cultivars. Ampelometry dates back more than 120 years, when Ravaz [10] introduced the main principles of the methodology. During the past decades, several refinements and specifications were suggested concerning the sampling [78,129], methodology [39,40,65,75] and data evaluation [61,117], which makes measurements faster and more accurate with higher discriminative power.

The leaf is available in almost the entire vegetation period and metric evaluations are objective according to the quantitative and continuous numerical values. An advanced benefit is the possibility of computerization [130], which was widely applied in the past decades. Among the methods, traditional ampelometry (measurements of linear and angular traits) and geometric ampelometry—both landmark-based and outline analysis—are possible to be performed on digital platforms, and this makes the measurements less time consuming and more reproducible, since samples can be stored as digital information.

Recently, morphometric variability between and within species, cultivars, clones, and clone candidates were explored and the traits that have discriminative power were highlighted. These traits are not necessarily the same in all investigations. The reasons for this are the different sample sets, and those external factors that influence the morphometric traits. According to related studies, both biotic (e.g., virus infection) [123] and abiotic factors (e.g., row orientation, irrigation) [92], and vineyard management practices (e.g., pruning level/budload) modify the ampelometric features [128]. The climatic condition is also a significant factor, as year-to-year studies showed large differences [102].

Investigations of the traits had a noticeable development. In the beginning of the 20th century, traditional morphometric (linear and angular) traits were described, and this was later supplemented with the calculation of ratios and categorization. Precursory studies on the position of certain landmarks in two-dimensional space were described in the 1940s. This approach was later improved with digital image analysis and multivariate statistical methods. This resulted in the landmark-based geometric morphometric evaluation of the leaf based on the cartesian coordinate analysis of certain biometric points on the leaf lamina. In the beginning of the investigations, landmarks were selected on the branching points of veins, and the tip of the lobes and teeth [76]. Later, not only the length of the veins, but also the width were evaluated according to further landmarks [77]. Meanwhile, other methods, such as outline analysis, were included in ampelometry, which were based more on the shape of the samples than on the size traits.

With the spread of new phenotyping and statistical evaluation methods, not only the grapevine leaf, but also, for example, the bunch [131], berry [132] and seed [133] would provide further possibilities to evaluate the diversity of the cultivars, help identification and explore the effect of the different environmental factors.

## 8. Conclusions

In 1902, Louis Ravaz [10] introduced a completely new methodology in the ampelographic investigations aimed to describe quantitative traits and improve the traditional characterizations. During the past 120 years, variously developed methods included digital image analysis and a multivariate statistical approach to clear synonymies, discriminate species and cultivars, and describe the natural variability within and between species, cultivars, and clones. In this review, we introduced the historical development of the methods and the most important innovations in ampelometry.

## Figures and Tables

**Figure 1 plants-12-00452-f001:**
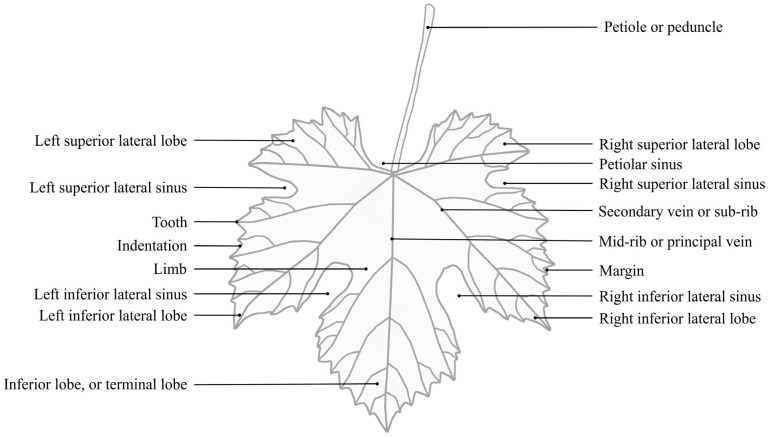
Grapevine leaf (according to Mazade [9]).

**Figure 2 plants-12-00452-f002:**
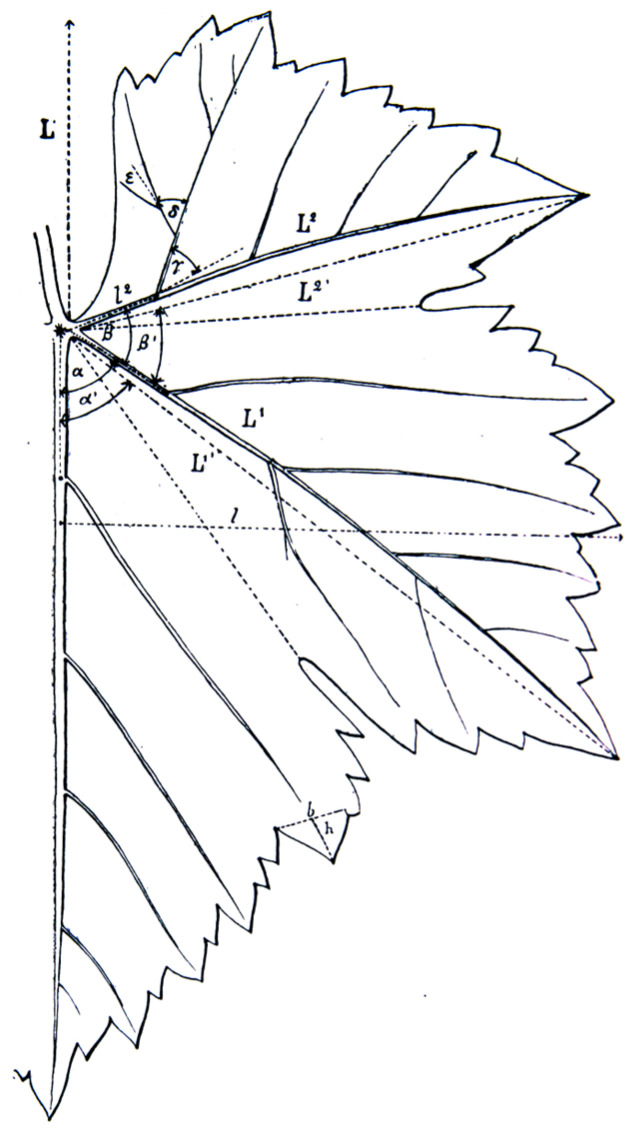
Leaf venation by Ravaz (1902) [10].

**Figure 4 plants-12-00452-f004:**
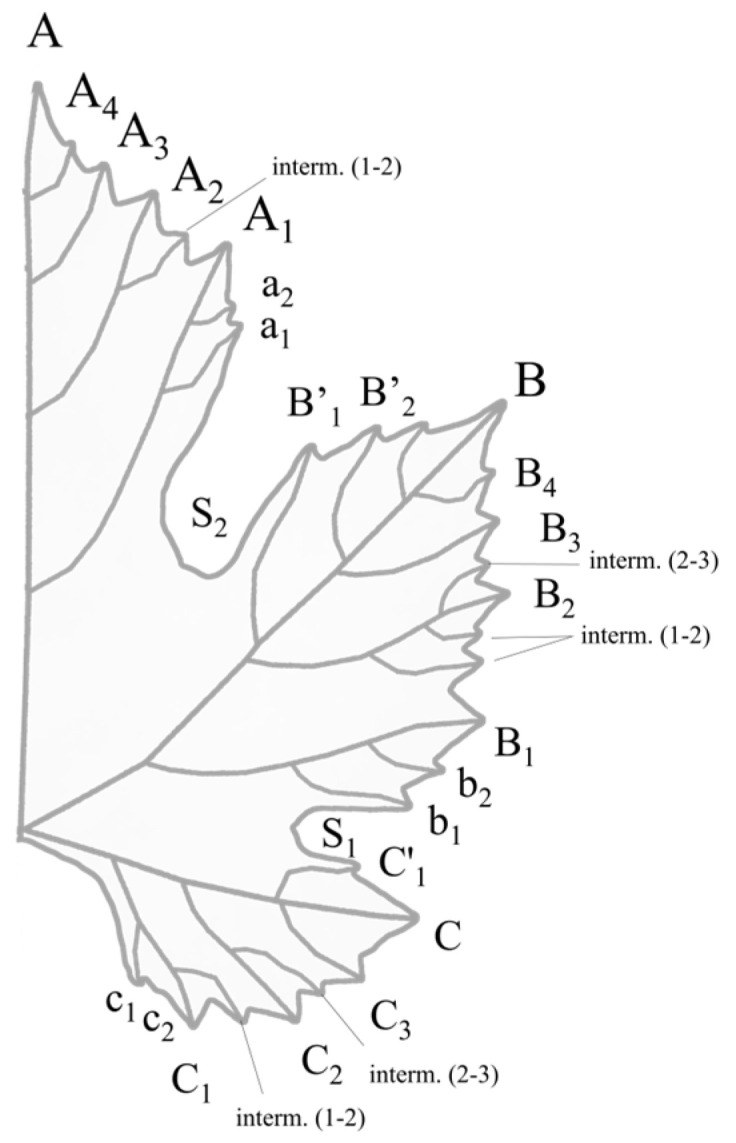
Ampelometric reference points suggested by Rodrigues [67] for evaluation of the leaf morphological diversity.

**Figure 5 plants-12-00452-f005:**
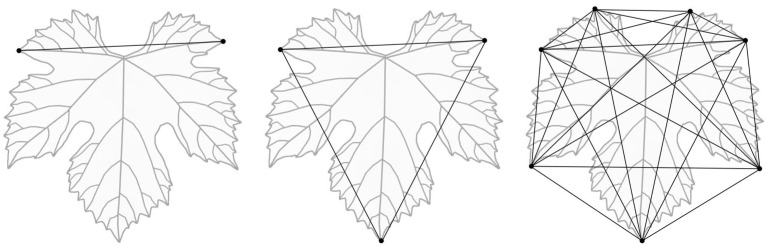
Calculated linear traits according to two, three and seven morphometric landmarks along the leaf boundary.

**Figure 7 plants-12-00452-f007:**
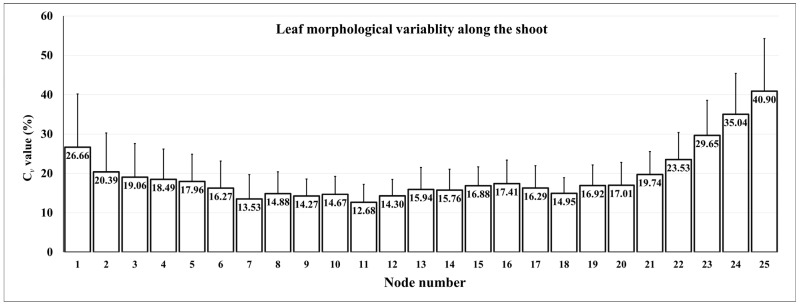
Grapevine (*Vitis vinifera* L.) leaves show noticeable variability along the shoot axis. This phenomenon was investigated according to 52 morphometric traits and those coefficients of variability (C*_v_*) at each node in Bodor et al. [78] (with permission from the publisher), who found that samples collected from the 11th node have the lowest variability among 10 shoots.

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
