# Peer review of "A Review of Ampelometry: Morphometric Characterization of the Grape (Vitis spp.) Leaf"

_plants, 2023, doi:10.3390/plants12030452_

Round 1
Reviewer 1 Report
The manuscript reports the importance of ampelometry in the identification and discrimination of grapevine biotypes. Amperometry has contributed significantly in the past to the ampelographic studies. In our days the codes of OIV are used as a complete tool including molecular markers. Ampelography represents the first step of grapevine identification, however sometimes it is not able to differentiate grapevine varieties. DNA genotyping by molecular markers allows identifying varieties despite of plant phenotype and changes in morphology. However this article may have an historical importance, and may serve as a complementary method to the modern ones.
On the other hand, there are many flaws in the text of the manuscript and incorrect sentences, some of which I have noted. So, the manuscript should be extensively revised by the authors.

Reviewer 2 Report
Thank you for submitting your manuscript for consideration. I found your review to be well-written, thorough and interesting. Well done.
Please take a moment to review the few comments I have provided below and address them thoroughly in the resubmission process.
Line 32: véraison - please correct throughout.
Line 46 - 47: The statement of rootstock cultivars having mainly male flowers, although reasonably true, I take issue with the qualifying statement about having no characterization importance. There are notable female and sterile rootstocks. I would encourage a revision of this sentence.
Reviewer 3 Report
Dear Author,
After peer review of the manuscript entitled “A review of ampelometry: morphometric characterization of the grape (Vitis spp.) leaf”, I suggest the MS needs minor revision:
In line 96, should cioutat be Chasselas cioutat?
In line 194 remove de from “the de characterization”
In line208 change from “short (up to about 75 mm)” to “short (about 105 mm)”
In line 235 legend should be changed from “Table 2” to “Table 1”
In line 288 “landmarks to 5” should be changes to “landmarks to 7” as shown in Figure 5
In line 338 is mentioned Figure 8 but I suppose should be Figure 7
In line 508 change “is are “ with “are”
The sentence from line 554 should be revised or clarified in particular the concept “measurements time consuming”
Round 2
Reviewer 1 Report
Although important corrections have been made to the text of the paper, however, there are still flaws, some of which I have noted in the text. So, the manuscript should be improved by the authors

Author Response
Budapest, 12th of January, 2023
Editorial Board of Plants
Response letter for Reviewer #1
Dear Reviewer,
On behalf of the co-authors, I would like to thank your time and helpful revision of our paper entitled “A review of ampelometry: morphometric characterization of the grape (Vitis spp.) leaf”.
We followed your valuable suggestions and corrected the manuscript. We simplified many of the sentences and made them clearer.
We hope you will find our corrections and the revised manuscript acceptable.
Thank you again for your helpful contribution.
Your sincerely,
Péter Bodor-Pesti,
corresponding author
